# Improved Survival and Symptom Relief Following Palliative Cerebrospinal Fluid Diversion for Leptomeningeal Disease from Brain Cancers: A Case Series and Systematic Review

**DOI:** 10.3390/cancers17020292

**Published:** 2025-01-17

**Authors:** Adela Wu, James Zhou, Stephen Skirboll

**Affiliations:** 1Department of Neurosurgery, Stanford University School of Medicine, Stanford, CA 94304, USA; skirboll@stanford.edu; 2Kaiser Permanente Los Angeles Medical Center, Elk Grove, CA 95757, USA; 3Section of Neurosurgery, VA Palo Alto Health Care System, Palo Alto, CA 93404, USA

**Keywords:** cancer, hydrocephalus, leptomeningeal disease, palliative, shunt

## Abstract

Leptomeningeal disease from cancer indicates widespread or advanced cancer and can lead to obstructive hydrocephalus, for which palliative cerebrospinal fluid diversion may be indicated to relieve symptoms, such as headaches and nausea. The authors of this study performed a review of published clinical outcomes for patients diagnosed with hydrocephalus from leptomeningeal disease. In addition, an analysis of a single-institution case series of adult patients was conducted to evaluate surgical outcomes, including complication and symptom relief rates. Symptom relief and improved survival can be achieved with permanent cerebrospinal fluid diversion for palliative purposes with low complication rates. Hence, this patient population may benefit from surgery to alleviate symptoms from hydrocephalus despite the poor prognoses development of leptomeningeal disease portends. Still, prospective studies are needed in the future to evaluate specific needs and impacts on quality of life for these patients.

## 1. Introduction

Leptomeningeal disease (LMD) is a devastating indication of widespread or advanced cancer that involves the meninges or has disseminated through the cerebrospinal fluid (CSF) system. LMD can be diagnosed at initial presentation for 2–12% of metastatic solid cancer cases, and rare etiologies, such as primary leptomeningeal melanoma, derived from melanocytes of the leptomeninges, have also been reported [1,2]. Presenting characteristics include cranial nerve palsies, weakness, sensation loss, as well as headache, nausea, vomiting, and obtundation from symptomatic hydrocephalus. Patients with neoplastic LMD have a poor prognosis, with a median survival of 1–4 months if untreated [1,3]. However, treating hydrocephalus with palliative CSF diversion, such as endoscopic third ventriculostomy (ETV) or placement of ventriculoperitoneal (VP) or lumboperitoneal shunts, can be effective for symptom relief. Despite potential complications of infection, revision, and peritoneal seeding of disease, patients diagnosed with LMD could benefit from undergoing palliative surgery.

We analyzed a cohort of adult patients with cancer from our institution who developed LMD and symptomatic obstructive hydrocephalus. We evaluated various outcomes, including rates of symptom relief and of complications as well as survival data following palliative CSF diversion surgery. The systematic review summarizes published series of adult and pediatric patients diagnosed with symptomatic hydrocephalus from neoplastic LMD and assesses the rates of surgery for CSF diversion. We evaluated various outcomes, including the rates of symptom relief and of complications as well as survival data following palliative shunt or ETV surgery.

## 2. Materials and Methods

### 2.1. Case Series

#### 2.1.1. Data Source and Patient Selection

All adult (age > 18) patients who were diagnosed with cancer, LMD, and obstructive hydrocephalus along with a date of death documented in the Stanford Health Care electronic medical record system between 1 January 2012 and 31 December 2022 were included for further analysis. Collected metrics included patient characteristics (age, sex, and racial/ethnic background), primary cancer type, date of primary cancer diagnosis, date of LMD diagnosis, date of CSF diversion surgery if applicable, date of death, and surgical outcomes (alleviated symptoms, surgical complications, and time between surgery and death). This study was approved by the Stanford University Institutional Review Board, and patient consent was waived based on the study design.

#### 2.1.2. Statistical Analysis

Categorical variables characterizing individual patients were analyzed using either the χ^2^ test or Fisher’s exact test. Fisher’s exact test was used when expected values fell below a numerical cut-off of 5. Continuous data were analyzed using the Wilcoxon rank-sum test. Survival analysis was calculated with the Kaplan–Meier analysis using the Cox proportional hazards model to evaluate differences between groups. All statistical testing was performed in R version 4.3.1.

### 2.2. Systematic Review

The literature search was developed by defining the population, intervention, comparison, outcomes, timing, and study design (PICOTS) questions and inclusion and exclusion criteria (Figure 1). The criteria, outcome measures, and search strategy were defined prior to analysis. The register number is CRD42022349960 ID.

#### 2.2.1. Search Strategy

The protocol was designed in accordance with preferred reporting items for systematic reviews and meta-analyses (PRISMA) guidelines [4]. The review included quantitative studies of pediatric and adult patients with primary or systemic cancers and tumors involving LMD and symptomatic hydrocephalus treated with CSF diversion, with at least five subjects, and which were written in English. The exclusion criteria included gray literature, non-systematic reviews, commentaries, and case reports [5]. In collaboration with a university librarian, we deployed a comprehensive search strategy within the PubMed, Embase, and Web of Science databases from inception until 20 August 2022. Synonymous words for key search terms were included to maintain the high inclusivity of the initial search.

#### 2.2.2. Study Selection

Duplicates were removed and study eligibility was determined using the PICOTS question and inclusion and exclusion criteria (Figure 1). The titles and abstracts, followed by article full texts, were independently screened by three study authors (A.W., J.Z., S.S.), with group discussion used to adjudicate discrepancies.

#### 2.2.3. Data Extraction and Analysis

We extracted key features from the eligible studies, including design, patient population characteristics, diagnosis of LMD, and surgical outcomes involving palliation and survival.

#### 2.2.4. Quality Assessment

Two authors (A.W. and J.Z.) independently used the Newcastle–Ottawa Scale to assess the quality of the studies [6]. The studies were graded as very low, low, medium, or high quality based on selection criteria, risk of bias, and overall study design [7]. No studies were excluded from analysis due to quality.

## 3. Results

### 3.1. Case Series

#### 3.1.1. Patient Characteristics

Fifty patients were identified, of whom thirty underwent surgery for CSF diversion due to symptomatic hydrocephalus. For the surgical group, the median age was 58.41 ± 14.38 years, while the median age of the non-surgical patient cohort was 57.83 ± 14.50 years (Table 1). There was no significant difference in patient age between the two groups (*p* = 0.426). The proportion of male patients in the non-surgical group (55.0%) was significantly greater than the surgical group percentage (23.3%) (*p* = 0.022). There were no significant differences in the breakdown of racial/ethnic backgrounds between the two cohorts. Within the surgical group, nine patients (30%) received Ommaya reservoirs; all others had VP shunts with programmable valves and antibiotic-impregnated catheters. The most common primary diagnosis was lung cancer in both the non-surgical (40.0%) and surgical (66.7%) groups. Nineteen patients from the surgical group underwent CT chest–abdomen–pelvis imaging for evaluation of their primary disease around the time of CSF diversion surgery, with no evidence of peritoneal disease seen on imaging.

Within the group of patients who did not undergo CSF diversion surgery, most were ultimately deemed not to be appropriate surgical candidates based on procedural risks outweighing potential benefits. Other reasons included unfavorably high protein count in CSF and patients’ refusal to undergo surgery.

#### 3.1.2. Outcomes of Palliative Cerebrospinal Fluid Diversion

The most common specified symptoms of obstructive hydrocephalus were headaches and nausea/vomiting for both patient groups. Other presenting symptoms were cranial neuropathies, gait imbalance, altered mental status, and bladder or bowel incontinence. Within the surgical cohort, headaches (36.7%) and nausea/vomiting (33.3%) were most commonly alleviated with CSF diversion. Overall, 22 patients (73%) reported some level of symptom relief or had improved neurologic status postoperatively. One complication occurred during the immediate postoperative period, which was a patient who unfortunately developed hemiparesis from the surgery.

There was a significant increase in the median time between LMD diagnosis and death for the surgical cohort (6.62 ± 6.00 versus 1.28 ± 4.29 months, *p* < 0.001) (Table 1). The median time between surgery and death was 2.75 ± 3.75 months. In the Kaplan–Meier survival curve analysis, there was a significant difference between the two groups, and there was a significantly lower likelihood of death with CSF diversion surgery (hazard ratio [HR] 0.402, 95% confidence interval [CI] 0.221–0.730, *p* = 0.002) (Figure 2).

### 3.2. Systematic Review

The search yielded 2440 studies for screening, of which 647 were duplicated studies and immediately excluded (Figure 3). Based on title and abstract review, we then excluded 1641 studies. A further 132 records were excluded based on full-text review. After inclusion of three additional studies based on a manual check of references, a final total of 23 studies was reviewed, with total 2895 patients, of which 995 had LMD on presentation.

#### 3.2.1. Study Characteristics

Studies were published between 1993 and 2022. These studies spanned eight countries (n = 8 United States [8,9,10,11,12,13,14,15]; n = 4 Korea [16,17,18,19]; n = 4 Japan [20,21,22,23]; n = 3 Canada [24,25,26]; n = 1 United Kingdom [27]; n = 1 Switzerland [28]; n = 1 France [29]; n = 1 Taiwan [30]). Twenty-two studies were retrospective cohort studies, and one was a case–control study in design. The most commonly reported data related to LMD included presenting symptoms and rates of LMD diagnosed on presentation. The most commonly reported surgical outcomes were rates of permanent CSF diversion, median overall survival, complication rates, and rates of hydrocephalus-related symptom relief.

#### 3.2.2. Patient Characteristics

Across all the studies, the patients’ diagnoses were primary brain tumors (n = 14) [8,10,13,14,15,17,18,20,24,25,26,27,28,29], such as medulloblastoma and glioma, as well as solid and hematologic malignancies, including lymphoma, lung, and breast cancers (n = 11) [9,11,12,15,17,19,21,22,23,30]. Both pediatric and adult patients were represented, with the median age at diagnosis ranging from 2.5 to 64.5 years old. The overall average percentage of male patients from the 18 studies that reported the statistic was 47.8%.

#### 3.2.3. Symptomatic Hydrocephalus

Nine studies reported diagnosing LMD and tumor dissemination through CSF cytology evaluation and/or evidence of leptomeningeal spread on neuraxis imaging [8,9,11,14,16,18,20,24,27]. Overall, LMD was diagnosed based on positive CSF cytology (9–82.6%) or MRI enhancement (56–100%). Studies that evaluated LMD with both cytology and MRI had higher percentages of LMD diagnosis based on imaging; for example, 100% of LMD was diagnosed in brain and spine MRI compared to 37.5% of cases resulting with positive CSF findings in Kwon et al.’s case series [18]. Among the studies that described the proportion of LMD diagnosed on presentation, 4.6–100% of patients had evidence of LMD on disease presentation [8,9,10,11,12,13,14,15,16,17,18,19,21,22,23,25,26,27,28,29,30].

The presenting symptoms of hydrocephalus commonly included headache, nausea/vomiting, seizures, cranial nerve palsies, obtundation, hemiparesis, and ataxia. Not all patients diagnosed with LMD develop symptomatic hydrocephalus. Ten studies reported a range of 13.8–80% of patients diagnosed with both neoplastic LMD and symptoms of increased intracranial pressure [8,9,11,13,14,16,19,26,28,29].

#### 3.2.4. Outcomes of Palliative Cerebrospinal Fluid Diversion

Fourteen studies reported proportions of patients with a disseminated tumor and hydrocephalus who underwent surgery for permanent CSF diversion (range: 9.9–100%) [8,9,11,12,16,17,18,21,22,23,25,26,27,30]. In total, 561 patients across all studies underwent CSF diversion for hydrocephalus from tumor dissemination. Specific data from studies that reported on types of surgical procedures included 5 endoscopic third ventriculostomies (ETV), 389 ventriculoperitoneal (VP) shunts, and 61 lumboperitoneal shunts. Of note, Lin et al. described the management of symptomatic hydrocephalus with a VP shunt equipped with an on–off valve on the reservoir [11]. Other methods of CSF diversion were placement of Ommaya reservoirs or Torkildsen shunts.

Among the patients who underwent surgery for palliative CSF diversion, in five studies that reported data on improvement in features of symptomatic hydrocephalus, 106 people (68%; range: 50–100%) experienced rapid symptom relief [11,12,17,22,23]. Murakami et al. investigated the impact of surgery on specific symptoms of increased intracranial pressure, including 88% of patients who reported relief from headaches and 67% of patients who had improved cognitive function and nausea relief [22]. One study documented Karnofsky Performance Scores (KPS) pre- and postoperatively; the median KPS improved from 30 prior to palliative CSF diversion to 70 postoperatively [23].

Postoperative complication rates were 0–37.7%, as reported by seven studies, with no cases of extraneural metastases via shunt. Other complications included shunt malfunction. Median overall survival for patients diagnosed with LMD ranged from 2.1 to 17.1 months in 14 studies. While Jung et al. reported a median survival after shunt surgery of 5.7 months, Omuro et al. described an additional median survival after shunt surgery of 2 months beyond a median survival after LMD diagnosis of 4 months [12,16]. The cohort in Bander et al.’s study had a median survival after shunt surgery of 2.43 months [9].

#### 3.2.5. Study Quality

Using the Newcastle–Ottawa Scale, 1 study was found to be of moderate quality and 22 were of low quality (Table 2). None were found to be of high quality.

## 4. Discussion

In our case series of 50 patients presenting with obstructive hydrocephalus from LMD secondary to cancer, we found that patients who underwent palliative surgery for CSF diversion had high rates of symptom relief, minimal surgical complications, and significantly longer survival than their counterparts. We also performed a systematic review of 23 studies, investigating rates and outcomes of palliative CSF diversion for symptomatic hydrocephalus resulting from neoplastic LMD. There were no randomized controlled trials that met our eligibility criteria, so future prospective studies regarding the impact of palliative CSF diversion for meningeal carcinomatosis are warranted.

LMD can develop from numerous tumor etiologies and portends poor survival. While the majority of neoplastic LMD spreads from metastatic breast or lung cancer, there is a range of studies about patients diagnosed with both metastatic primary or secondary brain cancers and rare syndromes, such as primary leptomeningeal melanoma [31]. With advances in diagnostic evaluations as well as oncologic management, the overall incidence of LMD is increasing, highlighting the importance of addressing LMD’s clinical sequelae for patients despite its devastating prognosis [32]. The clinical manifestations of LMD vary among patients, as cerebral involvement can result in headaches, nausea/vomiting, seizures, cranial nerve palsies, and obtundation, among other neurologic deficits [33,34]. These symptoms are characteristic of hydrocephalus, a common LMD-related cause of mortality and morbidity, which can develop as a result of tumor cells interfering with the outflow of CSF through arachnoid granulations in addition to consequences of reactive inflammation from tumor infiltration [33].

The placement of shunts or other interventions for CSF diversion are considered palliative for patients suffering from LMD and symptomatic hydrocephalus, as these procedures do not reverse disease course. There are certain complications commonly associated with shunt placement regardless of the primary disease process, including wound healing issues and shunt infection or malfunction, in addition to the rare risk of malignant ascites or the peritoneal seeding of disease through shunt catheters for patients diagnosed with metastatic tumors [35,36,37,38]. Among the case series described in this review, Kim, Murakami, and Omuro et al. were the only studies that specifically recorded the rates of peritoneal seeding and found no patients with this complication [12,17,22]. However, surgery for CSF diversion can effectively alleviate numerous debilitating symptoms of increased intracranial pressure. The few studies in our review that reported data on survival following shunt surgery cite up to an average of 5.7 months, which is nevertheless a substantial amount of time for meaningful symptom relief [9,12,16]. There are not enough studies with standardized methodology to compare the utility of ETV versus ventriculo- or lumboperitoneal shunts, however. As for the nonsurgical management of symptomatic hydrocephalus arising from the leptomeningeal spread of cancer, few options beyond anti-emetics and pain medications directed at symptoms are available, such as repeated lumbar punctures [39]. Otherwise, treatment for leptomeningeal disease that involves radiation, chemotherapy, and palliative steroids may impact CSF flow and alleviate some symptoms of hydrocephalus temporarily [16].

Beyond patient-reported relief from a finite sample of preoperative symptoms, the impact of palliative CSF diversion could potentially be better understood with additional measures of quality of life and functional status postoperatively. No recent studies specifically reported standardized metrics related to quality of life prior to or following CSF diversion. Some studies included KPS as an indication of functional status, with improvements in condition after palliative surgery seen in Yoshioka et al.’s cohort [9,23,30]. While the collection of data on patients’ functional status (i.e., Karnofsky Performance Status, ECOG Performance Status) was not standardized among our cohort either, understanding a global picture of impairment versus functional independence would be valuable additional information for not only cancer prognosis but potentially surgical candidacy and recovery as well [40].

The limitations of this study include the heterogeneity of the study design and the cohort size and type, such as the various tumor etiologies of neoplastic LMD, thereby restricting the generalizability of reported outcomes. The patients included in the case series were also assessed in a non-standardized manner due to different providers’ clinical practices. Therefore, certain data were not available for further analysis, including the Karnofsky Performance Scale and other factors pertaining to patients’ functional status. This systematic review also describes retrospective studies, which involve biases and confounding factors. Furthermore, this review is also limited by the nature and heterogeneous scope of outcomes, as studies do not uniformly report data, such as on rates of symptom relief or complications, for example.

## 5. Conclusions

A rare entity with a dismal prognosis, LMD and disseminated tumors can present as symptomatic hydrocephalus in patients with cancer. Symptom relief, including improved headaches and cognition, can be achieved with palliative surgery for permanent CSF diversion with relatively low complication rates and no incidence of extraneural metastases through shunt catheters. Further prospective studies are needed to further assess additional outcomes and needs as well as aspects of quality of life for this patient population.

## Figures and Tables

**Figure 1 cancers-17-00292-f001:**
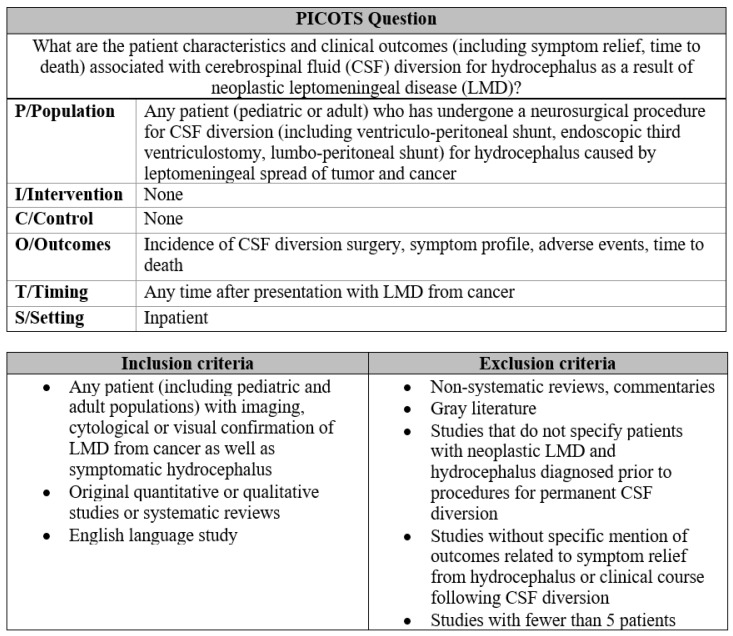
PICOTS question and inclusion and exclusion criteria.

**Figure 2 cancers-17-00292-f002:**
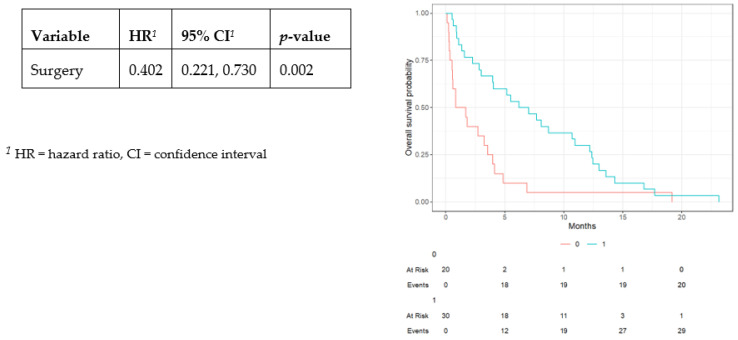
Kaplan–Meier survival curve for non-surgical and surgical patient groups with leptomeningeal disease.

**Figure 3 cancers-17-00292-f003:**
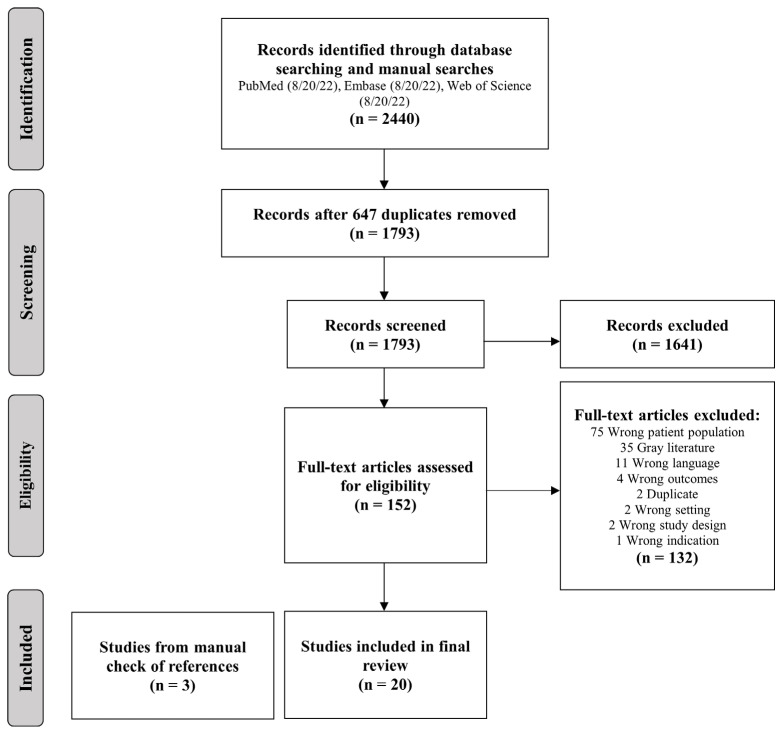
Flow diagram of the systematic review process.

**Table 1 cancers-17-00292-t001:** Patient characteristics in case series of patients who did and did not undergo surgery for cerebrospinal fluid diversion.

	No Surgical Diversion (n = 20, 40.0%)	Surgical Diversion (n = 30, 60.0%)	*p*-Value
Age at diagnosis (median ± SD)	57.83 ± 14.50	58.41 ± 14.38	0.426
Male	11 (55.0%)	7 (23.3%)	0.022
Race			
White	9 (45.0%)	14 (46.7%)	0.908
Asian	6 (30.0%)	11 (36.7%)	0.626
Other	5 (25.0%)	5 (16.7%)	0.494
Hispanic	3 (15.0%)	1 (3.3%)	
Primary cancer diagnosis			
Lung	8 (40.0%)	20 (66.7%)	0.063
Brain	3 (15.0%)	3 (10.0%)	0.672
Breast	4 (20.0%)	5 (16.7%)	1
GI	3 (15.0%)	1 (3.3%)	0.289
Other	2 (10.0%)	1 (3.3%)	0.556
Initial LMD symptoms			
Headache	6 (30.0%)	14 (46.7%)	0.239
Nausea/vomiting	7 (35.0%)	12 (40.0%)	0.721
Cranial neuropathies	2 (10.0%)	2 (6.7%)	1
Altered mental status	6 (30.0%)	10 (33.3%)	0.805
Gait	4 (20.0%)	5 (16.7%)	1
Incontinence	1 (5.0%)	2 (6.7%)	1
Miscellaneous	9 (45.0%)	9 (30.0%)	0.279
Symptoms alleviated			
Headache	N/A	11 (36.7%)	N/A
Nausea/vomiting	N/A	10 (33.3%)	N/A
Cranial neuropathies	N/A	1 (3.3%)	N/A
Altered mental status	N/A	3 (10.0%)	N/A
Gait	N/A	2 (6.7%)	N/A
Incontinence	N/A	0 (0.0%)	N/A
Miscellaneous	N/A	7 (23.3%)	N/A
Time between LMD and death (months ± SD)	1.28 ± 4.29	6.62 ± 6.00	<0.001
Time between surgery and death (months ± SD)	N/A	2.75 ± 3.75	N/A

**Table 2 cancers-17-00292-t002:** Summary of twenty-three eligible studies with >5 patients. CN: cranial nerve. CNS: central nervous system. CSF: cerebrospinal fluid. DIPG: diffuse infiltrative pontine glioma. ETV: endoscopic third ventriculostomy. GBM: glioblastoma. LMD: leptomeningeal disease. LP: lumboperitoneal. VP: ventriculoperitoneal.

Study ID (Country of Study)	Study Design; Study Quality [7]	Study Time Period	Inclusion Criteria	Sample Size	Pertinent Study Findings on Hydrocephalus Treatment	Survival
Allcutt et al., 1993 (Canada) [26]	Retrospective; Low	1961–1991	All pediatric patients with primary leptomeningeal melanoma at The Hospital for Sick Children with sufficient diagnostic information	8	% with permanent CSF diversion: 6 (75%) [4 LP shunts; 1 VP shunt; 1 Torkildsen shunt]	Median overall survival: 6 months
Andersen et al., 2019 (United States) [9]	Retrospective; Low	2001–2016	All adult patients with confirmed LMD from Grade 2–4 glioma at Memorial Sloan Kettering Cancer Center	188	% with permanent CSF diversion: 36 (23%) [36 VP shunts]	Median overall survival of LMD at presentation: 8.3 months
Bander et al., 2021 (United States) [10]	Retrospective; Low	2010–2019	All patients diagnosed with LMD and treated hydrocephalus at Memorial Sloan Kettering Cancer Center	190	% with permanent CSF diversion: 190 (100%) [189 VPS; 1 V-pleural shunt]	Median overall survival: 4.14 months; Median survival after shunt: 2.43 months
Castro et al., 2017 (United States) [11]	Retrospective; Low	2004–2014	All patients with surgically resected GBM at University of California at San Francisco	841	No specific data available	No specific data available
Fischer et al., 2014 (Switzerland) [30]	Retrospective; Low	2007–2011	All patients with completely resected and treated GBM	151	No specific data available	No specific data available
Jung et al., 2014 (Korea) [18]	Retrospective; Low	2005–2012	Adult patients with LMD from systemic solid tumors treated at Chonnam National University Hwasun Hospital	71	% with permanent CSF diversion: 7 (9.9%) [7 VP shunt]	Median survival after shunt surgery: 5.7 months; Median overall survival: 2.1 months
Kim et al., 2019 (Korea) [19]	Retrospective; Low	2002–2017	Patients with confirmed LMD and treated hydrocephalus at National Cancer Center in Korea	70	% with permanent CSF diversion: 100% [51 VP shunt; 19 LP shunt]	Median survival after LMD diagnosis: 8.7 months
Kirkman et al., 2018 (United Kingdom) [29]	Retrospective; Low	2003–2016	All pediatric patients with confirmed central nervous system (CNS) tumors with tumor dissemination evident on MRI	361	% with permanent CSF diversion among patients with LMD: 36 (67.9%)	No specific data available
Kwon et al., 2020 (Korea) [20]	Retrospective; Low	2004–2019	Patients with confirmed primary LMD from high-grade glioma (Grade III-IV)	9	% with permanent CSF diversion: 66.7% [4 VP shunt; 2 Ommaya]	Median survival: 263 days
Lee et al., 2011 (Korea) [21]	Retrospective; Low	2003–2010	Patients with diagnosed CNS metastases who underwent VP shunt	50	No specific data available	Median survival after LMD diagnosis: 3.5 months
Le Fournier et al., 2017 (France) [31]	Retrospective; Low	2005–2015	All pediatric patients with newly diagnosed posterior fossa tumor admitted to Angers University Hospital and Rennes University Hospital	29	No specific data available	No specific data available
Lin et al., 2011 (United States) [13]	Case-control; Moderate	2005–2009	All adult patients with LMD treated at Huntsman Cancer Institute or Brigham and Women’s Hospital/Dana-Farber Cancer Institute	24 (case); 24 (control)	% with permanent CSF diversion among cases: 24/24 (100%) [24 RO-VP shunt]	Median progression-free survival among cases: 14 weeks
Matsumoto et al., 2006 (Japan) [22]	Retrospective; Low	1979–2003	Patients with diagnosed germ cell tumor or medulloblastoma and hydrocephalus treated with percutaneous long-tunneled ventricular drainage (PLTVD) at single institution	13	No specific data available	No specific data available
Mitsuya et al., 2019 (Japan) [23]	Retrospective; Low	2008–2017	Patients with LMD from lung adenocarcinoma and hydrocephalus requiring treatment at Shizuoka Cancer Center	31	% with permanent CSF diversion: 100% [13 VP shunt; 19 LP shunt]	Median overall survival after LMD diagnosis: 4.5 months; Median overall survival after shunt surgery: 3.5 months
Murakami et al., 2018 (Japan) [24]	Retrospective; Low	2007–2016	Patients with palliative shunt placement for hydrocephalus and LMD not amenable for surgical resection at a single institution	11	% with permanent CSF diversion: 100% [8 VP shunt; 3 LP shunt]	Median survival after LMD diagnosis: 3.9 months; Median survival after shunt surgery: 3.3 months
Omuro et al., 2005 (United States) [14]	Retrospective; Low	1995–2003	All adult patients with diagnosed LMD (primary brain tumors excluded) and VP shunt treated at Memorial Sloan-Kettering Cancer Center	37	% with permanent CSF diversion: 100% [37 VP shunt]	Median survival after LMD diagnosis: 4 months; Median survival after shunt surgery: 2 months
Rennert et al., 2021 (United States) [15]	Retrospective; Low	2010–2020	Pediatric patients with ETV for hydrocephalus from primary brain tumors at Rady Children’s Hospital of San Diego	15	No specific data available	Median survival among patients with LMD: 2.5 months
Rinaldo et al., 2018 (United States) [16]	Retrospective; Low	2001–2016	Patients with Grade III or IV glioma with hydrocephalus treated by shunting at Mayo Clinic Rochester	41	No specific data available	No specific data available
Riva-Cambrin et al., 2009 (Canada) [27]	Retrospective; Low	1989–2003	Patients ≤ 17 years of age with newly diagnosed posterior fossa tumors at Hospital for Sick Children in training cohort; pediatric patients with posterior fossa tumors at British Columbia Children’s Hospital in validation cohort	Training cohort (343); validation cohort (111)	Training cohort: % with permanent CSF diversion: 15/107 (14.3%)	No specific data available
Sandberg et al., 2000 (United States) [17]	Retrospective; Low	1995–1998	Patients with LMD and Ommaya reservoirs and surgically treated hydrocephalus at Memorial Sloan-Kettering Cancer Center	107	No specific data available	Median survival after LMD diagnosis: 8.5 months
Schneider et al., 2015 (Canada) [28]	Retrospective; Low	1991–2013	Patients with medulloblastoma and hydrocephalus treated at The Hospital for Sick Children	130	% with permanent CSF diversion among patients with LMD/solid metastases: 12/28 (42.9%)	No specific data available
Su et al., 2022 (Taiwan) [32]	Retrospective; Low	2017–2020	Patients with LMD from lung cancer treated at a single institution	50	% with permanent CSF diversion: 40/50 (80%) [33 VP shunt; 7 LP shunt]	Median overall survival: 4.9 months
Yoshioka et al., 2021 (Japan) [25]	Retrospective; Low	2010–2019	Patients with diagnosed LMD and hydrocephalus who underwent treatment at Kindai University Hospital	14	% with permanent CSF diversion: 100% [5 VP shunt; 9 LP shunt]	Median overall survival: 3.7 months

## Data Availability

The original contributions presented in this study are included in the article. Further inquiries can be directed to the corresponding author.

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
