# Peer review of "Improved Survival and Symptom Relief Following Palliative Cerebrospinal Fluid Diversion for Leptomeningeal Disease from Brain Cancers: A Case Series and Systematic Review"

_cancers, 2025, doi:10.3390/cancers17020292_

Round 1
Reviewer 1 Report
Comments and Suggestions for Authors
This is an important topic for clinicians managing patients with disseminated cancer. It highlights the potential role of VP shunt implantation as a palliative procedures for those suffering from the painful effects of LMD. Although the results are convincing, the manuscript can benefit from a more thoughtful discussion. To make the discussion more meaningful, I suggest the following:
1. To include more details of the authors' experience with the shunt discussion, especially with regards to: type of VP shunt implant (?fixed pressure valve/ programmable valve, ?antibiotic-coated catheter), perioperative-related complications and time of survival from shunt insertion to demise. This part of the discussion should also emphasize on what are the specific risks associated with VP shunt insertion for this group of patients with terminal disease.
2. To include in the discussion other types of less invasive treatment for symptomatic LMD and rationale for the choice of a neurosurgical intervention instead. (eg. steroids, palliative radiotherapy, intermittent bedside lumbar puncture)
Author Response
|
Comment 1: To make the discussion more meaningful, I suggest the following: To include more details of the authors' experience with the shunt discussion, especially with regards to: type of VP shunt implant (?fixed pressure valve/ programmable valve, ?antibiotic-coated catheter), perioperative-related complications and time of survival from shunt insertion to demise.
|
|
Response 1: We appreciate your suggestions. Table 1 includes information about time between cerebrospinal fluid diversion surgery and date of death. One perioperative-related complication was noted in the cohort and described in the Results (page 5, lines 146-148). We have added details about the shunt implants to the Results (pages 4, lines 128-129). Revisions to the text are in red font color.
|
|
Comments 2: This part of the discussion should also emphasize on what are the specific risks associated with VP shunt insertion for this group of patients with terminal disease.
|
|
Response 2: Agreed. We included additional references addressing this point to the Discussion (page 14, lines 263-267). Revisions to the text are in red font color.
Comments 3: To include in the discussion other types of less invasive treatment for symptomatic LMD and rationale for the choice of a neurosurgical intervention instead. (eg. steroids, palliative radiotherapy, intermittent bedside lumbar puncture)
Response 3: We appreciate your insights and have added more information and references to the Discussion (page 14, lines 275-280).
|
Reviewer 2 Report
Comments and Suggestions for Authors
In this manuscript the authors present what can be considered one of the most difficult issue to face in neurosurgery, hydrocephalus secondary to cancer disseminaton to the meningeal envelope. This would make the study extremely interesting, especially for the number of cases included. unfortunately, some factors have not been highlighted correctly and I would like the authors to be more specific. First of all, it seems that the diagnosis of leptomeningeal dissemination in some cases was made only by CSF citology, which can be barely significant in patients where the presense of brain metastases has yet been ascertained. Although identification of pachymeningeal cancer dissemination can be extremely difficult even with high field MRI machines, why did not the author consider a CSF subtraction test or a Katzmann test? Considering that only 22 of the 30 patients undergoing shunting improved, they might have cleaned the slate beforehand, sparing yet compromised patients an useless procedure. Obviously, nothing can be said about the 20 that did not received the procedure. Why were these cases left untreated? Because hydrocephalus was seen at MRI but was not symptomatic? This might have well excluded patients that could have even a longer survival if treated preventively, before developing symptoms. And again, in these cases at least a CSF registration pressure test could have helped. My second concern is related to the types of cancers listed. As an example, gastrointestinal cancers are well known to raise perineal invasion before spreading further. Did the authors consider a VA shunting in these cases, due to the potential failure of VP procedures? I would ask the authors if they did run a full diagnostic (CT chest-abdomen pelvis) in the immediate proximity to surgery. My third concern is coming from my experience with these patients. It happens to find hydrocephalus and simultaneous meningeal thickening, evident even at a standard CT. Did the authors have similar cases? Did they put the shut the same? Ot did they operated in these cases to remove the mass effect? or nothing? Finally, it is quite interesting that the authors do not mention the KPS of these patients, something that is considered paramount before deciding for surgery in oncologic situations of disseminated disease. If they simply omitted it, please include the pre and post-operative KPS for operated and non operated patients. It would add an important point to the whole paper.
P.S. In the abstract it says : "leptomeningeal disease from brain cancer". This has to be changed in leptomeningeal disease from cancer"
Author Response
|
Comment 1: In this manuscript the authors present what can be considered one of the most difficult issue to face in neurosurgery, hydrocephalus secondary to cancer disseminaton to the meningeal envelope. This would make the study extremely interesting, especially for the number of cases included. unfortunately, some factors have not been highlighted correctly and I would like the authors to be more specific.
|
|
Response 1: Thank you for your time and attention in reviewing our manuscript. |
|
Comments 2: First of all, it seems that the diagnosis of leptomeningeal dissemination in some cases was made only by CSF citology, which can be barely significant in patients where the presense of brain metastases has yet been ascertained. Although identification of pachymeningeal cancer dissemination can be extremely difficult even with high field MRI machines, why did not the author consider a CSF subtraction test or a Katzmann test?
|
|
Response 2: Thank you for your insights. At our institution where our cases were collected, it was not standard practice for these patients to undergo testing such as digital subtraction myelography or a Katzman test to diagnose hydrocephalus and leptomeningeal disease.
Comments 3: Considering that only 22 of the 30 patients undergoing shunting improved, they might have cleaned the slate beforehand, sparing yet compromised patients an useless procedure. Obviously, nothing can be said about the 20 that did not received the procedure. Why were these cases left untreated? Because hydrocephalus was seen at MRI but was not symptomatic? This might have well excluded patients that could have even a longer survival if treated preventively, before developing symptoms. And again, in these cases at least a CSF registration pressure test could have helped.
Response 3: We appreciate your comments. CSF pressure registration tests are not regularly used at our institution. The vast majority of patients in our cohort who did not undergo CSF diversion surgery were consulted by the inpatient neurosurgical team and ultimately deemed not to be surgical candidates in that the risk of shunt placement surgery overrode potential benefits. Other reasons for not surgically treating other patients included unfavorably high protein count in CSF and patients declining the procedure (page 4, lines 134-137). Revisions are in red font color in the manuscript.
Comments 4: My second concern is related to the types of cancers listed. As an example, gastrointestinal cancers are well known to raise perineal invasion before spreading further. Did the authors consider a VA shunting in these cases, due to the potential failure of VP procedures? I would ask the authors if they did run a full diagnostic (CT chest-abdomen pelvis) in the immediate proximity to surgery.
|
Response 4:
Of the patients who received CSF diversion surgery in our cohort, one had a primary diagnosis of colon/gastrointestinal cancer, and a ventriculoperitoneal shunt was placed for palliation. Nineteen patients (19/30; 63%) underwent CT chest-abdomen-pelvis around the time of surgery with no evidence of peritoneal disease seen on imaging (page 4, lines 131-133). Revisions are in red font color in the manuscript.
Comments 5:
My third concern is coming from my experience with these patients. It happens to find hydrocephalus and simultaneous meningeal thickening, evident even at a standard CT. Did the authors have similar cases? Did they put the shut the same? Ot did they operated in these cases to remove the mass effect? or nothing?
Response 5:
Thank you for sharing your clinical experience. The patients in our institutional cohort were diagnosed with leptomeningeal spread on cytology or MRI and did not routinely undergo CT head imaging. No patients in the case series had a history of brain surgery to remove lesions causing mass effect after diagnosis of leptomeningeal disease.
Comments 6:
Finally, it is quite interesting that the authors do not mention the KPS of these patients, something that is considered paramount before deciding for surgery in oncologic situations of disseminated disease. If they simply omitted it, please include the pre and post-operative KPS for operated and non operated patients. It would add an important point to the whole paper.
Response 6:
Thank you for raising this important point. We wanted to include data on patients’ functional status pre- and postoperatively. However, based on different providers’ clinical practices, patients in the case series had not been assessed in a standardized manner. As such, we have included this point as a limitation in the Discussion (page 14, lines 294-297). Revisions are in red font color in the manuscript.
Comments 7:
P.S. In the abstract it says : "leptomeningeal disease from brain cancer". This has to be changed in leptomeningeal disease from cancer"
Response 7:
We updated this sentence in the Abstract (page 1, line 22).
Round 2
Reviewer 1 Report
Comments and Suggestions for Authors
No further comments.